# Effect of Paclobutrazol on the Physiology and Biochemistry of *Ophiopogon japonicus*

**Zezhou Zhang** [1,2,†], **Ruixing Li** [3,†], **Deyong Chen** [4], **Jieyin Chen** [2], **Ouli Xiao** [1,2], **Zhiqiang Kong** [2] and **Xiaofeng Dai** [1,2,*]

1    Feed Research Institute, Chinese Academy of Agricultural Sciences, Beijing 100081, China; zhangzezhou7689@163.com (Z.Z.); xiaoouli123@163.com (O.X.)
2    State Key Laboratory for Biology of Plant Diseases and Insect Pests, Institute of Plant Protection, Chinese Academy of Agricultural Sciences, Beijing 100193, China; chenjieyin@caas.cn (J.C.); kongzhiqiang@caas.cn (Z.K.)
3    Haidian District Market Supervision and Administration Bureau, Beijing 100080, China; liruixing06@163.com
4    College of Life Sciences, Tarim University, Alar 843300, China; chendy0824@163.com
*    Correspondence: daixiaofeng_caas@126.com
†    These authors contributed equally to this work.

**Abstract:** *Ophiopogon japonicus* is a commonly used Chinese medicine with multiple pharmacological effects. To increase the yield of *O. japonicus*, paclobutrazol is widely used during the cultivation, and residues of paclobutrazol cause undesired side effects of *O. japonicus*. In this study, the effect of different concentrations of paclobutrazol on *O. japonicus* was investigated, and the final residual amount of paclobutrazol in the plant sample was determined by ultra-performance liquid chromatography-tandem mass spectrometry (UPLC-MS/MS); cell morphology was observed by transmission electron microscopy. The inhibitory effect of paclobutrazol on plant height and the stimulatory effect on root elongation were concentration-dependent from 0.6 to 11.3 g/L, reaching a maximum of about 28% and 67%, respectively. However, when the concentration was 22.5 g/L, these effects were significantly weakened, and the same trend was observed for the tuber root weight. Paclobutrazol caused the cell wall of *O. japonicus* to thicken, making the cells smaller and more densely arranged. Paclobutrazol also inhibited bacterial growth, irrespective of the concentration. Considering the residual concentration after application and the effects on growth, the application of 1.3 g/L or 2.8 g/L paclobutrazol can increase the accumulation of effective ingredients while promoting production, reducing application costs, and maximizing farmers' profit.

**Keywords:** paclobutrazol; *Ophiopogon japonicus*; physiological effects; biochemical effects

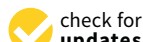



## 1. Introduction

*Ophiopogon japonicus* (*O. japonicus*) is a well-known traditional Chinese medicine with a long history of usage and various pharmacological activities that have significant immediate effects [1]. The definition of *O. japonicus* in the Chinese Pharmacopoeia 2020 is "*O. japonicus*, the dried tuber roots of Liliaceae *Ophiopogon japonicus* (L.f) Ker-Gawl" [2]. Meanwhile, *O. japonicus* has the functions of nourishing yin and body fluid, nourishing the lungs and clearing the heart [1]. Steroidal saponins and homoisoflavonoids are the two main active substances in *O. japonicus* from the phytochemistry studies; more than 100 steroidal saponins, 50 homoisoflavonoids, 10 bioactive polysaccharides, and some other kinds of organic acids and phenylpropanoids were isolated and identified [3], which have a variety of pharmacological activities, such as against myocardial ischemia [4], and they exhibit antioxidant [5], anti-inflammatory [6], and anti-diabetic [7] effects. Currently on the market, many health foods for clearing and moisturizing throats contain *O. japonicus*, such as Zhigancao Decoction [8], Maimendong Decoction [9], and Zengye Decoction [10].

In recent years, *O. japonicus* has been widely planted in Sichuan and accounting for more than 95% of total *O. japonicus* output in China [11]. To increase the yield and shorten

the growth cycle of Ophiopogonis Radix, the plant growth regulator paclobutrazol is widely used [12]. In the genuine producing areas of Ophiopogonis Radix (Santai, Sichuan, China), the amount of paclobutrazol (wettable powder 15%) used per 667 m$^2$ is about 3–6 kg [13]; converted into concentration, it is about 5.6–11.3 g/L. The application concentration of paclobutrazol should be kept within a certain range. Too high concentration of paclobutrazol has certain phytotoxic effects on crops, and this conclusion has been confirmed in corn, broccoli, and other crops [14–16]. In China, plant growth regulators are managed as pesticides and must be registered before being used. Nevertheless, their application to most traditional Chinese medicine is still at a stage of lack of registration. Up to January 2021, there are only two pesticides registered on *O. japonicus*: the herbicide tribenuron-methyl and the fungicide tebuconazole [17], but there are no relevant regulations for paclobutrazol. Plant growth regulators can regulate the physiological processes at different stages in the growth and development of traditional Chinese medicine; they also regulate the secondary metabolism, thereby affecting the content of effective active substances in traditional-Chinese medicine [18]. Paclobutrazol ((2RS, 3RS)-1-(4-chlorophenyl)-4, 4-dimethyl-2-(1H-1, 2, 4-trizol-1-yl)-pentan-3-ol) is one of the members of the triazole family having a growth-regulating property. The growth-regulating properties of PBZ are mediated by changes in the levels of important plant hormones including the gibberellins, abscisic acid, and cytokinins [19]. As a result, paclobutrazol inhibits the growth of the above-ground part of *O. japonicus*, promotes the growth of the roots of the underground part, and increases the yield of *O. japonicas* [12]. However, in the absence of supervision, abuse of paclobutrazol will cause changes in the effective components of *O. japonicus*. Studies have found that paclobutrazol excessive use reduced the content of Ophiopogonin D, Ophiopogonin C, and Ophiopogonanone C; in addition, paclobutrazol poses a threat to the growth environment of *O. japonicus*, and paclobutrazol was also detected even in the planting soil and water [13]. Therefore, the excessive use of paclobutrazol has double residual hazards to traditional Chinese medicine and the cultivation environment, causing serious harm to human health.

So far, most studies have been focused on the analysis of pesticide residues in traditional Chinese medicines. There are few studies on the effects of plant growth regulators on the physiology and biochemistry of traditional Chinese medicine plants. Here, we explored the effects of paclobutrazol on the physiological and biochemical effects of *O. japonicus*. Seven different applied concentrations of paclobutrazol were set up for field trials, and the residues of paclobutrazol and the changes in active substances content were determined with the UPLC-MS/MS. Meanwhile, plant height, root length, root diameter etc. of *O. japonicus* were analyzed.

## 2. Materials and Methods

### 2.1. Chemicals and Reagents

Paclobutrazol with purity exceeding 98.0% was purchased from Dr. Ehrenstorfer (LGC Standards; Augsburg, Germany), and the standard stock solutions (100 mg/L) of paclobutrazol were prepared in LC-grade acetonitrile. The standard working solutions of paclobutrazol at 5, 10, 50, 100, 200, 800, and 1000 μg/L were prepared in pure acetonitrile from the stock solution by serial dilution. All the solutions were wrapped with aluminum foil and stored in a refrigerator at −20 °C prior to analysis. The wettable powder of 15% effective content paclobutrazol was purchased from Jiangsu Kesheng Group Co., Ltd, (Nanjing, China). Methylophiopogonone A, Methylophiopogonanone A, Methylophiopogonanone B, Ophiopogonanone C, Ophiopogonanone E, Ophiopogonanone A, Ophiopogonin D, Ophiopogonin D', Ophiopogonin B, and Ophiopogonin Ra, with purities greater than 98%, were purchased from Yuanye Biological Technology Co., Ltd. (Shanghai, China). Acetonitrile and formic acid (HPLC grade) were purchased from Thermo Fisher Scientific (Fisher, NJ, USA). Sodium chloride (NaCl) and anhydrous magnesium sulfate (MgSO$_4$) were purchased from Beijing Chemical and Reagent (Beijing, China). Octadecylsilyl (C18) was obtained from Agela Technologies Co., Ltd. (Tianjin, China). Glutaraldehyde

was purchased from Shanghai Jingke Chemical Technology Co., Ltd. (Shanghai, China). MP Biomedicals Sodium Phosphate Buffer was purchased from Beijing Zeping Technology Co., Ltd. (Beijing, China). Osmium Tetroxide (4% in Water) was purchased from Sinopharm Chemical Reagent Beijing Co., Ltd. (Beijing, China). Ultra-pure water was obtained using a Milli-Q water purification system (Millipore, Bedford, MA, USA). An Embed-812 Resin Embedding Kit was purchased from Hyde Venture Biotechnology Co., Ltd. (Beijing, China).

### 2.2. Field Trial Plot Selection and Field Application Method

The test base is located in Santai County (Sichuan, China) (31°23'23" N, 104°49'40" E, 453 m). It is a subtropical humid monsoon climate zone with an average annual temperature of 16–17 °C. The annual average sunshine hours ≥1260 h, rainfall is 850–900 mm, frost-free period ≥275 d, and the soil type is sandy soil. There is no paclobutrazol applied in the early stage. A total of seven paclobutrazol spray concentrations of 0 (CK group), 0.6, 1.3, 2.8, 5.6, 11.3, and 22.5 g/L were set in the experiment. About 0 g, 7.5 g, 17.5 g, 37.5 g, 75 g, 150 g, and 300 g of paclobutrazol (15% wettable powder) were dissolved in 2 L of water to prepare paclobutrazol solutions; then, the different concentrations were uniformly sprayed on the leaves of plants with small sprayers. Each concentration was used in three randomized experimental districts, each district was 4 $m^2$, and the interval protection behavior was 1 m. Field experiments were started at 20 October 2019; at harvest time (8 March 2020), tuberous roots of *O. japonicas* from the control group and treated groups (twenty-one samples in total) were collected for evaluation. The samples of *O. japonicus* were pulverized and homogenized after drying and then sealed in Ziploc bags. All samples were stored at 4 °C prior to analysis.

### 2.3. The Final Residue of Paclobutrazol in O. japonicus

The method of determination of paclobutrazol residue in *O. japonicus* sample refers to Li et al. [20] with slight modification. Homogenized samples (2 g) were placed into a 50 mL centrifuge tube and mixed with 10 mL water. Furthermore, 6 mL acetonitrile containing 0.75% acetic acid was added to the mixture and ultrasonically extracted for 15 min at room temperature. The sample was centrifuged at 8000 rpm for 5 min under 10 °C. Thereafter, 1.0 mL of the upper organic layer was introduced into a new Teflon centrifuge tube containing 150 mg $MgSO_4$ and 25 mg C18. The solution was mixed and vortexed for 1 min; then, it was centrifuged at 11,000 rpm for 5 min. Finally, the supernatant was filtered through a 0.22 μm nylon organic membrane and transferred to the injection vial prior for UHPLC-MS/MS analysis.

A 1290 Infinity UHPLC system coupled with a 6495A Triple Quadrupole mass spectrometer (UHPLC-QQQ-MS) (Agilent Technologies, Wilmington, DE, USA) equipped with a degasser, a binary pump, and an electrospray ionization source (AJS ESI), with multi-reaction monitoring for detection was used to obtain the best response and highest sensitivity. The column used was an Agilent Zorbax SB-C18 column of 50 mm × 2.1 mm, 1.8 μm (Agilent Zorbax Eclipse). The mobile phases were water containing 0.01% formic acid (phase A) and pure acetonitrile (phase B). Furthermore, the gradient elution was used as follows: 0–0.5 min, 5% B; 0.5–5 min, 5–99% B; 5–6 min, 99% B; 6.1–7 min, 5% B; finally, the mobile phase remained for 1 min under initial conditions to rebalance the system before the next injection, with a flow rate of 0.3 mL/min. Then, 2 μL of sample or standard solution was injected into the column. Paclobutrazol was determined by MRM with a positive ion. For the mass spectrometric analysis, nitrogen was supplied as the nebulizer and collision gas. The ion source parameters were set as follows: capillary voltage was 3.5 kV for the positive mode, source temperature of 150 °C, desolvation temperature of 325 °C, sheath gas (argon) flow of 11 L/h, and drying gas (nitrogen) flow of 10 L/h. The retention time of paclobutrazol was 9.301 min, the parent ion was 294, the daughter ion was 125/70, and the collision energy was 20/15, respectively.

### 2.4. Effect of Paclobutrazol on the Appearance and Yield of O. japonicus

The effect of paclobutrazol on the appearance and yield of *O. japonicus* was investigated with plant height, root length, root diameters, and 100-seed weight. For the appearance of *O. japonicus*, one plant was randomly selected from each concentration, and their photos were taken with a black background [21]. Fifteen plants were also randomly selected at each concentration, and their heights from the ground part were measured and recorded as the plant height (PLHT). Three tuber roots were randomly selected for each concentration, their photos were taken with a black background, and their diameters and lengths were measured with a Vernier calliper. For the yield of *O. japonicus*, one hundred tuber roots were randomly selected from each concentration, and their total weights were measured and recorded as the 100-seed weight (100 SW). Each group of experiments was repeated three times, and the average value was taken as the actual measured value.

### 2.5. Method of Transmission Electron Microscope

After being cut into pieces, *O. japonicus* was forthwith placed in 3% glutaraldehyde for fixation at room temperature (25 °C) for 24 h and then fixed at 4 °C for at least another 24 h. The fixed samples were rinsed twice with 0.1 M phosphate buffer, each time for 10 min; then, they were fixed with 1% osmium tetroxide for at least 2 h and then rinsed twice with 0.1 M phosphate buffer. After that, they were dehydrated with an ethanol and acetone gradient, then osmosized with an Embed-812 Resin Embedding Kit. All the samples were penetrated, embedded, sliced, stained, and attached to a nickel mesh. Sample observation and image acquisition were performed using an H-7650 transmission electron microscope and 382 charge-coupled device camera, respectively, with a working voltage of 80 kV [22].

### 2.6. Quantitative Detection of Saponins and Flavonoids in O. japonicus

Steroidal saponins and homoisoflavonoids are the main active substances of *O. japonicus* [23]. We selected Ophiopogonin D [24], Ophiopogonin D' [25], Ophiopogonin Ra with anti-cancer activity, Methylophiopogonanone A [26], Methylophiopogonanone B [27], which can improve cardiovascular disease function, Ophiopogonanone C, and Ophiopogonanone E with anti-inflammatory activity [6] as the examined substances.

The extraction method of *O. japonicus* adopts the pre-treatment method of Xie et al. [28]. According to the actual situation, the extraction time, extraction solvent volume, and solvent ratio are shortened: about 30 mL of 90% methanol was added to 1 g of *O. japonicus* powder, ultrasonically extracted for 45 min, centrifuged at 8000 rpm for 5 min, and 1 mL of supernatant was added to the injection vial.

Quantitative detection of saponins and flavonoids was similar to that previously described in Section 2.4. The mobile phases were water containing 0.01% formic acid (phase A) and pure acetonitrile (phase B). The gradient elution used was as follows: 0–0.1 min, 5% B; 0.1–1 min, 5–50% B; 1–5 min, 50–75% B; 5–6 min, 75–80% B; 6–7 min, 80% B; 7–7.1 min, 80–5% B; 7.1–8 min, 5% B; finally, the mobile phase remained for 1 min under initial conditions to rebalance the system before the next injection at a flow rate of 0.3 mL/min. Then, 2 μL of sample or standard solution was injected into the column. For the mass spectrometric analysis, nitrogen was supplied as the nebulizer and collision gas. The ion source parameters were set as follows: capillary voltage was 3.5 kV for the negative mode, source temperature of 150 °C, desolation temperature of 325 °C, sheath gas (argon) flow of 11 L/h, and drying gas (nitrogen) flow of 10 L/h. The MS/MS parameters for each analysis are shown in Table 1.

### 2.7. Data Processing

The data obtained from the experiments were processed with Microsoft Office Excel 2016 software. Origin 2018 was used for the drawings, and the SPSS 26 data processing system was used to analyze statistical significance. Data were presented as mean $\pm$ SD (mean $\pm$ standard deviation). The differential significance between the mean of CK and treatment groups was evaluated with one-way analysis of variance (ANOVA) followed

by a paired t-test and Dunnett's t-test correction. Significant difference was defined at the level of $p < 0.01$.

**Table 1.** LC-MS/MS parameters of the phenolic active substances' quantitative analyses.

| Compound | RT (min) | Precursor Ion [i] (m/z) | Product Ion [q] (m/z) | IM | FV (V) | CE (V) |
|---|---|---|---|---|---|---|
| Ophiopogonin D | 2.766 | 899.5 | 721.5/575.3 | ESI- | 300 | 33/40 |
| Ophiopogonin D′ | 2.759 | 899.5 | 721.5 | ESI- | 320 | 35 |
| Ophiopogonin Ra | 2.013 | 753.3 | 607.3/246.8 | ESI- | 320 | 30/30 |
| Methylophiopogonanone A | 3.449 | 341 | 205.9/178 | ESI- | 100 | 25/30 |
| Methylophiopogonanone B | 3.736 | 327.1 | 206/178 | ESI- | 150 | 25/35 |
| Ophiopogonanone C | 4.748 | 355.1 | 193/164 | ESI- | 200 | 33/38 |
| Ophiopogonanone E | 2.039 | 359 | 344/329 | ESI- | 100 | 20/25 |

RT: retention time; [q]: for quantification; [i]: for identification; IM: ionization mode; FV: fragmentor voltage; CE: collision energy.

## 3. Results and Discussion

### 3.1. The Final Residue of Paclobutrazol in O. japonicus

Currently, paclobutrazol is not registered for *O. japonicus* and there is no current regulation regarding its residue limit, and this caused many farmers to misunderstand the use of paclobutrazol in *O. japonicus*. They thought that more paclobutrazol could bring more benefits, so it was easy to use paclobutrazol heavily. The results of previous studies have shown that the detection rate of paclobutrazol in *O. japonicus* is extremely high, and the amount of paclobutrazol in most plants exceeds the MRL in vegetables and fruits specified in GB 2763-2019 [20].

Through field trials, the final residual of different concentrations of paclobutrazol in *O. japonicus* was tested and analyzed. The experimental results are shown in Figure 1. When 0.6 g/L, 1.3 g/L, 2.8 g/L, 5.6 g/L, 11.3 g/L, and 22.5 g/L paclobutrazol were applied, the final residues were 0.08 mg/kg, 0.12 mg/kg, 0.35 mg/kg, 0.47 mg/kg, 0.73 mg/kg, and 1.35 mg/kg, respectively. As the maximum residue limit (MRL) of paclobutrazol in *O. japonicus* was not specified, compared with the MRL (0.5 mg/kg) in vegetables and fruits specified in GB 2763-2019, when the application concentration is 11.3 g/L and 22.5 g/L, the residual amount of paclobutrazol exceeds 0.5 mg/kg. There are research reports that 100% of *O. japonicus* samples collected in Sichuan have detected paclobutrazol residues [29]. According to the report by Li et al. [20], the residual amount of paclobutrazol in *O. japonicus* is 4.18 to 783.73 µg/kg, and most of the paclobutrazol residue exceeds 0.5 mg/kg. Based on those research results, it is common to use paclobutrazol during the growth process of *O. japonicus* in Sichuan province.

### 3.2. The Effect of Paclobutrazol on the Appearance and Yield of O. japonicus

#### 3.2.1. Effect of Paclobutrazol on Plant Height of *O. japonicus*

Figure 2 shows the effect of paclobutrazol application on the aerial parts of *O. japonicus*. Paclobutrazol had an obvious inhibitory effect on the height of *O. japonicus* plants in the range of 0.6 to 2.8 g/L, as the concentration increases, the plant height has an obvious downward trend. The specific results are shown in Figure 3. Without paclobutrazol application, the average PLHT in the control group was 35.56 cm, and even the application of 0.6 g/L of paclobutrazol had a significant inhibitory effect on the PLHT; the average PLHT was 33.82 cm. In the range of 0.6 to 11.3 g/L, as the concentration of paclobutrazol increased, its inhibitory effect became more obvious. When the application concentration was 11.3 g/L, the inhibitory effect reached a maximum, and the average PLHT was 25.43 cm, which was suppressed by 28.18% compared with the control group. However, when a high concentration of paclobutrazol (22.5 g/L) was applied, the inhibitory effect was weakened, and the average PLHT was 28.29 cm. In addition, the PLHT of *O. japonicus* between the application of 2.8 and 5.6 g/L paclobutrazol showed no significant difference, while the other groups all showed a significant difference. Paclobutrazol inhibits the growth of plants by inhibiting

the synthesis of sterols, prerequisite compounds for the synthesis of gibberellin, as well as inhibiting the synthesis of the plant endogenous hormone gibberellin [30]. The results are consistent with the findings of our previous field survey and monitoring data reported by Amani et al. [31], which indicates that paclobutrazol is a plant growth retardant that suppresses olive tree growth length while promoting the fruit set.

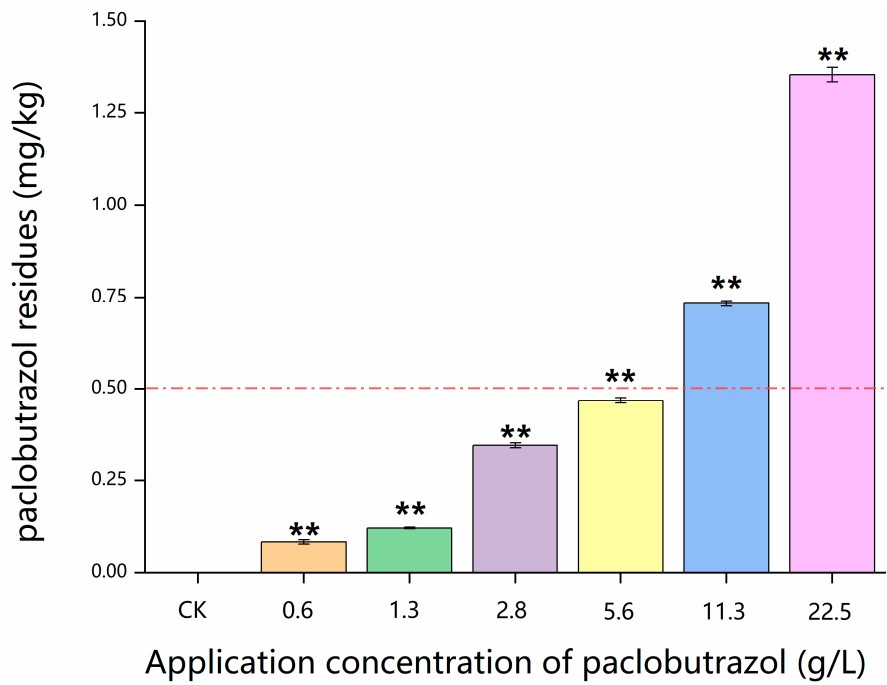

**Figure 1.** The final residues of paclobutrazol in *O. japonicus* after spraying different concentrations of paclobutrazol. Data are mean ± SD, *n* = three; ** *p* < 0.01, versus CK group (one-way ANOVA with Tukey's multiple comparison test). Vegetables and fruits MRL = 0.5 mg/kg in GB 2763-2019.

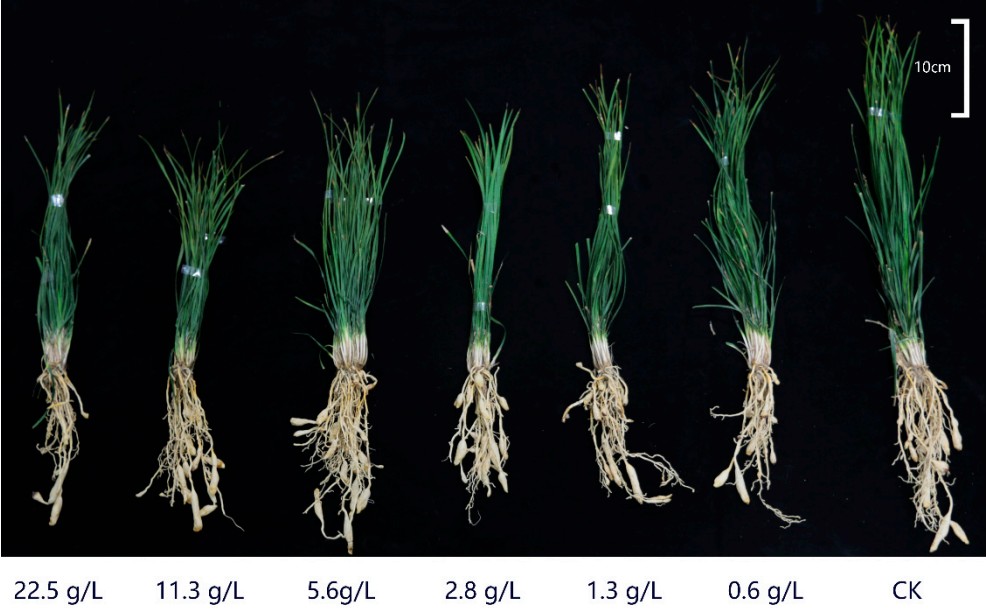

**Figure 2.** Comparison of whole *O. japonicus* plants after spraying different concentrations of paclobutrazol.

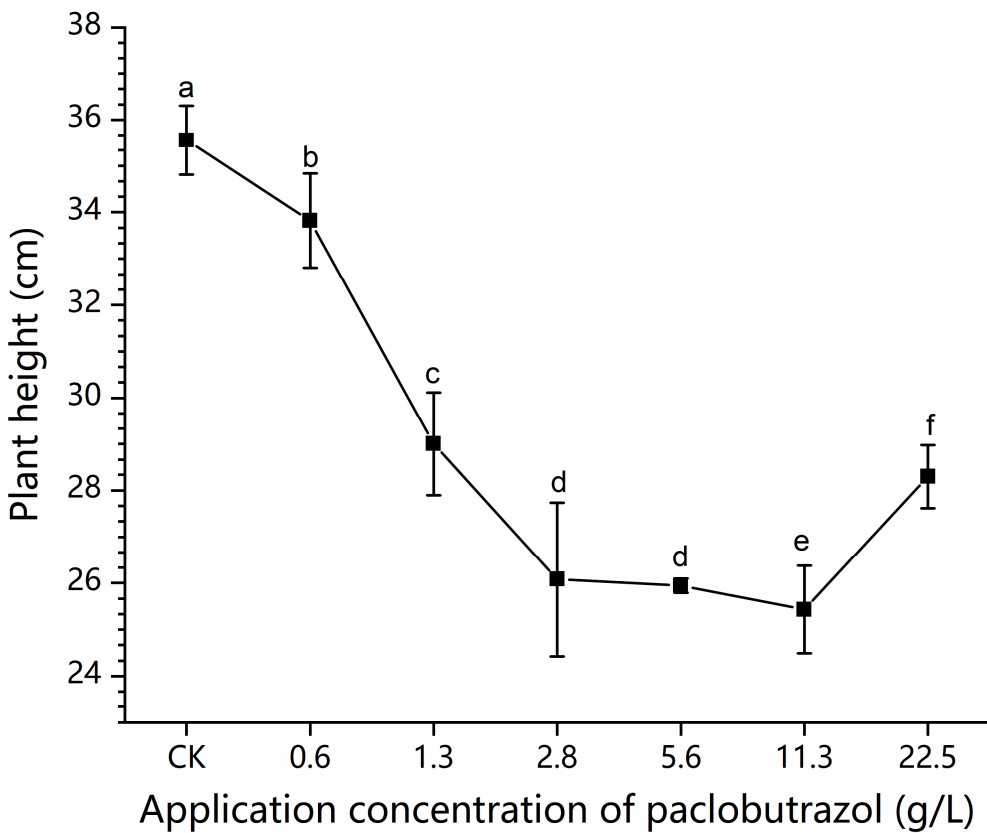

**Figure 3.** Effects of different concentrations of paclobutrazol on the plant height of *O. japonicus*. Different letters (a–f) represent significant differences ($p \leq 0.01$) across treatments according to Tukey's SD test. Vertical bars represent standard errors.

3.2.2. Effect of Paclobutrazol Treatment on the Diameter and Length of *O. japonicus* Root

Figure 4A shows the effect of paclobutrazol on the length and diameter of *O. japonicus* tuber roots.

Compared with the tuber roots in Figure 2, it can be seen that the groups with a high tuber root index have different tuber roots density. For example, the tuber roots of *O. japonicus* in the 5.6 g/L group are denser than in the 11.3 g/L group and only second to the CK group, but its tuber root index is less than 11.3 groups. This shows that it is not rigorous to compare specific index parameters from the whole plant. For tuber roots indexes, specific index analysis of specific parts is required.

It can be seen that in the range of 0.6–11.3 g/L, paclobutrazol has an obvious promotion effect on the length of *O. japonicus* tuber roots. The specific effect is shown in Figure 4B. The average length of *O. japonicus* in the CK group was 23.56 mm. At low concentration of 0.6 g/L, paclobutrazol could significantly promote the growth of *O. japonicus*, with an average root length of 28.07 mm. It can be seen that higher concentrations of paclobutrazol have different promoting effects on root length, but it is not positively correlated. When paclobutrazol is applied at 0.6 to 11.3 g/L, its effect on the growth of *O. japonicus* tuber roots is more obvious as the application concentration increases; with the promotion effect reaching a maximum at 11.3 g/L, the average length of *O. japonicus* roots can reach 39.56 mm, which is 67% longer than that without paclobutrazol treatment. When a high concentration of paclobutrazol (22.5 g/L) is applied, it has a reduced effect on length growth, with an average root length of 31.39 mm. Although it has a significant increase compared with the CK group, its growth effect is similar to that of the 1.3 g/L group and far less than the 1.5, 3, and 11.3 g/L groups.

For the tuber root diameter, the average diameter of *O. japonicus* was 8.02 mm. As shown in Figure 4B, although the application of paclobutrazol at different concentrations

has a significant promoting effect on tuber root diameter, it is different from the effect on the length of the tuber root. Not all concentrations promoted an increase in diameter. There was no significant difference between the concentrations of 0.6 g/L and 1.3 g/L on root tuber diameter, as well as the concentrations of 5.6 g/L and 11.3 g/L. The average diameter of *O. japonicus* tuber root reached a maximum when the concentration is 11.3 g/L, with an average diameter of 8.55 mm. Similar to the trend of tuber root length, when 22.5 g/L of paclobutrazol was applied, its diameter did not continue to increase but decreased instead, with an average diameter of only 8.39 mm. As shown in Figure 4A, the root size with 22.5 g/L of paclobutrazol is significantly smaller than that treated with a concentration of 11.3 g/L. When considering this result regarding the high concentration of paclobutrazol (22.5 g/L) leading to a change in the growth trend of tuber roots, we speculate that it is due to the high concentration of paclobutrazol causing phytotoxicity to *O. japonicus*; a high concentration of paclobutrazol did not promote but rather inhibited the production of cytokinins [19], which is similar to the study of Yeshitela [32].

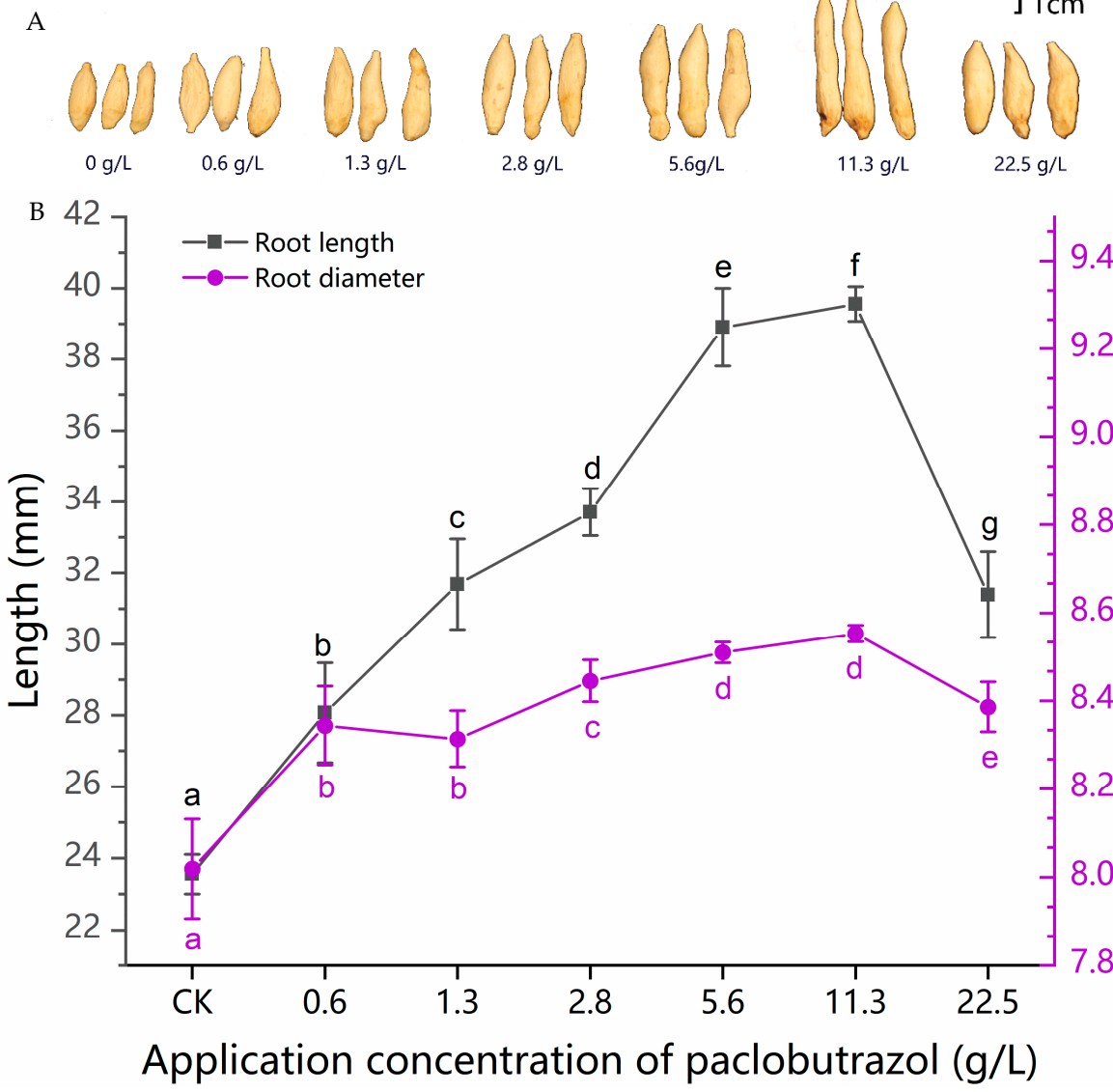

**Figure 4.** Overall comparison of Ophiopogon roots after application of paclobutrazol at different concentrations: (**A**) root diameter and root length (**B**). Different letters (a–g) represent significant differences ($p \leq 0.01$) across treatments according to Tukey's SD test. Vertical bars represent standard errors.

### 3.2.3. Effect of Paclobutrazol on the Root Weight of *O. japonicus*

One hundred nutrient tuber roots of *O. japonicus* were selected and weighed; then the total weights were calculated, and all the operations were repeated three times. The root weight of *O. japonicus* was determined and shown in Figure 5. Through the results, we can see that the application of paclobutrazol increase the weight of *O. japonicus*. The average 100 SW of the CK group was 49.45 g. The application of paclobutrazol at 11.3 g/L increased root weight the most, with an average 100 SW of 73.74 g, which is an increase of 49.12%. Among them, there was no significant difference in the 100 SW after the application of 0.6 g/L and 1.3 g/L of paclobutrazol, with an average weight of 61.17 g and 61.70 g, respectively. There was also no significant difference in the 100 SW after the application of 2.8 g/L and 5.6 g/L of paclobutrazol, with an average weight of 69.61 g and 69.15 g, respectively. From the results, it can be seen that a high concentration of paclobutrazol does not always improve the weight of *O. japonicus*. Although the application of 22.5 g/L paclobutrazol resulted in an increased weight, its promotion degree was obviously lower than that of the 11.3 g/L treatment, where the average 100 SW was 68.96 g. The application of 11.3 g/L of paclobutrazol increased the root weight of *O. japonicus* the most; however, this concentration led to a higher residual amount of paclobutrazol in the final *O. japonicus* samples. Therefore, when the paclobutrazol concentration is in the range of 0.6–5.6 g/L, it can ensure the promotional effect on *O. japonicus* tuber root growth and reduce the amount of paclobutrazol residue.

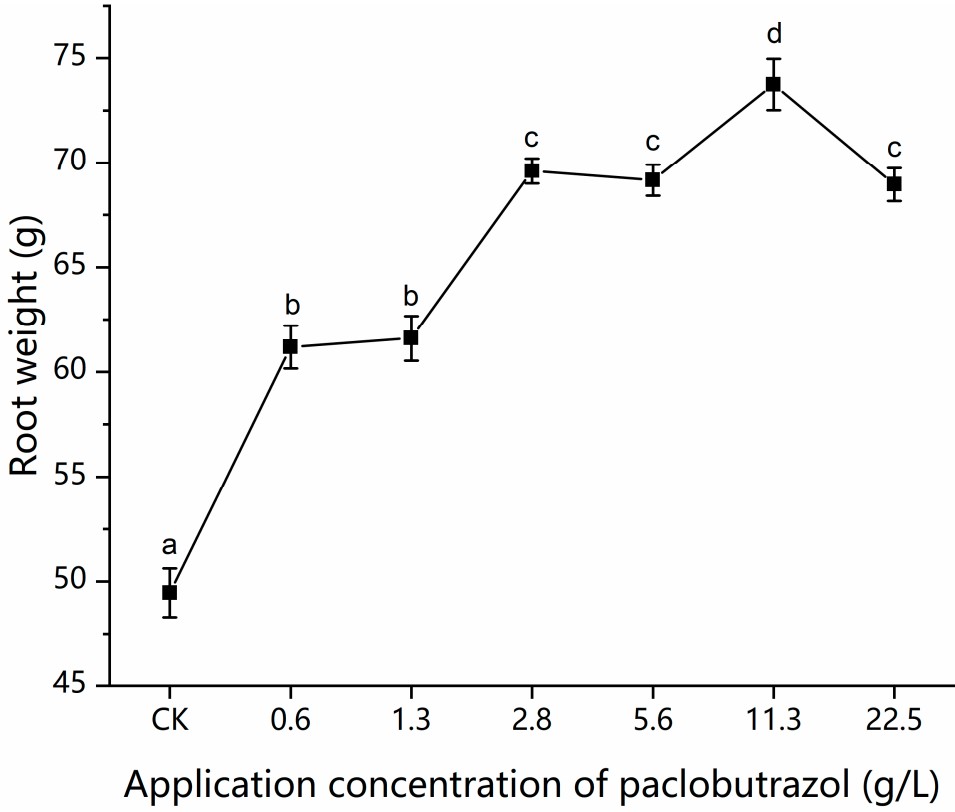

**Figure 5.** Effects of paclobutrazol concentration on root weight of *O. japonicus.* Different letters (a–d) represent significant differences ($p \leq 0.01$) across treatments according to Tukey's SD test; Vertical bars represent standard errors.

### 3.3. Effect of Paclobutrazol on the Cell Morphology of *O. japonicus*

According to the actual situation, we selected four groups (including the CK group) of paclobutrazol treatment groups with significant differences to analyze the results. The effect of paclobutrazol on the cell morphology of *O. japonicus* is shown in Figure 6. The results show that a low concentration of paclobutrazol can cause cell wall thickening, which

decreased with an increase in concentration (Figure 6A). At the same time, the cells became smaller and more tightly arranged (Figure 6B). A high concentration of paclobutrazol can cause the loss of organelles in the tuber root cells of *O. japonicus*. It can be seen from Figure 6C that compared with the CK group, the paclobutrazol treatment group has a loss of organelles.

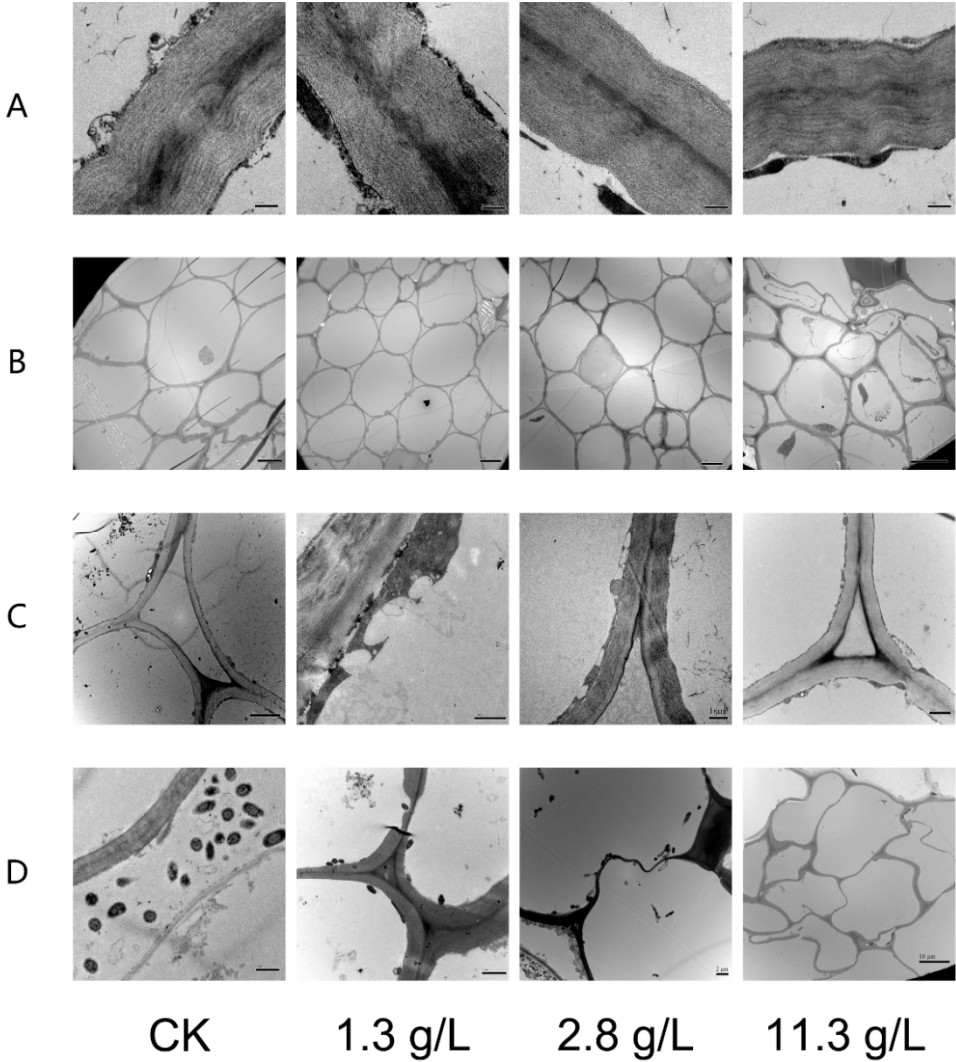

**Figure 6.** Effects of paclobutrazol at different concentrations on cells of *O. japonicus*. Effects on the cell wall (**A**), cell size and morphology (**B**), organelles (**C**), and number of bacteria (**D**).

At the same time, we found that paclobutrazol had a good inhibitory effect on the growth of bacteria since there were more bacteria in the cells of the CK group. As the concentration of paclobutrazol increased, the amount of bacteria decreased; there were almost no bacteria at high concentrations (Figure 6D). That may due to the bactericidal properties of paclobutrazol, which can inhibit the growth of bacteria [33]. All these phenomena explain that the application of paclobutrazol has an effect on the root cells of *O. japonicus*. Similar conclusions were found in research with torch ginger [34] and soybean [35].

### 3.4. The Effect of Paclobutrazol on the Flavonoids and Saponins in Ophiopogon japonicus

*O. japonicus* is a Chinese medicinal material; its active substances have medicinal value, so whether the application of paclobutrazol will affect the efficacy and quality of *O. japonicus* is worthy of attention [13]. Many studies found that Ophiopogonin D and Ophiopogonin D′ have anti-cancer activity [24,36,37]; Ophiopogonin Ra has a good cardiovascular protective function [38]. Some Chinese medicine decoctions contain Ophiopogon flavonoids, such as

Zhigancao decoction containing ophiopoganone [8] and Yi Guan Jian decoction containing methylophiopogonone [39]. Therefore, exploring the effect of paclobutrazol on the active substances in *O. japonicus* is helpful to rationally guide traditional Chinese medicine, to maximize the use of saponins and flavonoids in *O. japonicus* within a range of safe pesticide applications. The technique UHPLC-QQQ-MS was used to study the effects of different concentrations of paclobutrazol on Ophiopogonin D, Ophiopogonin D', and Ophiopogonin Ra in *O. japonicus*. The experimental results are shown in Figure 7. The contents of Ophiopogonin D, Ophiopogonin D', and Ophiopogonin Ra in *O. japonicus* in the CK group were 40.79 mg/kg, 42.61 mg/kg, and 22.17 mg/kg, respectively. When 0.3–5.6 g/L paclobutrazol is applied, it can increase the accumulation of these three saponins. The application of 1.3 g/L paclobutrazol can promote the maximum accumulation of saponins, which can increase Ophiopogonin D by 43% to 58.48 mg/kg, Ophiopogonin D' by 44% to 61.46 mg/kg, and Ophiopogonin Ra by 6% to 28.34 mg/kg. When the paclobutrazol concentration reached 11.3 g/L and 22.5 g/L, the accumulation of saponins was inhibited, and the concentrations of the three saponins were lower than those in the CK group. The results show that *O. japonicus* treated with 1.3 g/L of paclobutrazol has the most content of these three saponins, and the residual concentration of paclobutrazol is also too low to exceed 0.5 mg/kg. This discovery has a guiding role in the extraction and cultivation of natural *O. japonicus* saponins, and it reduces the medical cost of *O. japonicus*.

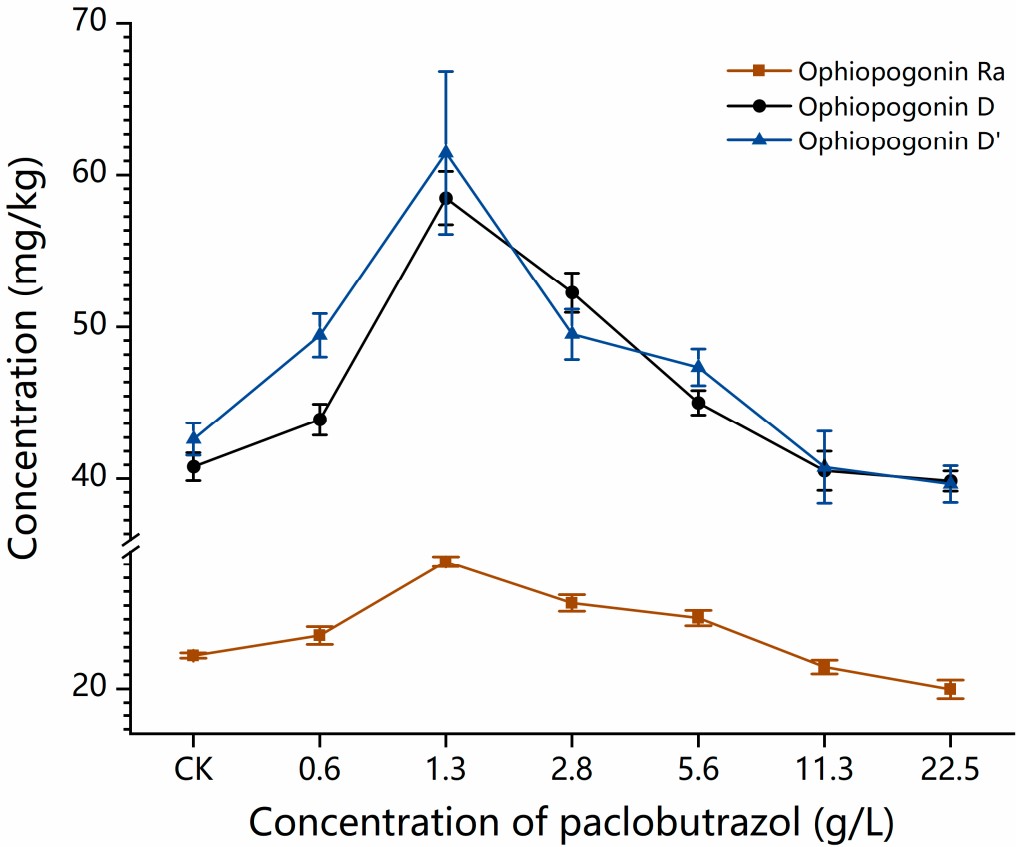

**Figure 7.** Effects of paclobutrazol at different concentrations on the contents of Ophiopogonin D, Ophiopogonin D', and Ophiopogonin Ra. Vertical bars represent standard errors.

Similarly, the effects of paclobutrazol on Methylophiopogonone A, Methylophiopogonone B, Ophiopoganone C, and Ophiopoganone E in *O. japonicus* were investigated. The experimental results are shown in Figure 8. The content of the four flavonoids of *O. japonicus* in the CK group were 151.84 mg/kg, 114.05 mg/kg, 18.86 mg/kg, and 19.33 mg/kg, respectively. Although different concentrations of paclobutrazol were effective, the trends of the various flavonoids were different, but they all reached the high-

est content at a concentration of 2.8 g/L paclobutrazol. This concentration increased Methylophiopogonone A by 19% to 180.15 mg/ kg, Methylophiopogonone B by 17% to 133.07 mg/kg, Ophiopoganone C by 50% to 29.69 mg/kg, and Ophiopoganone E by 52% to 29.49 mg/kg. Although the application of 22.5 g/L of paclobutrazol can also promote the accumulation of Methylophiopogonone A, Methylophiopogonone B, and Ophiopoganone C, it can lead to a higher residual amount of paclobutrazol; therefore, this dosage was not recommended. Although the optimal spraying concentration of flavonoids is different from that of saponins, considering the influence of PLHT, root size, and 100 SW, we believe that the spraying of paclobutrazol of 2.8 g/L can maximize the residual concentration when the residual concentration is much lower than 0.5 mg/kg. It can increase the output of *O. japonicus* while making the best use of the active substances saponins and flavonoids in *O. japonicus*.

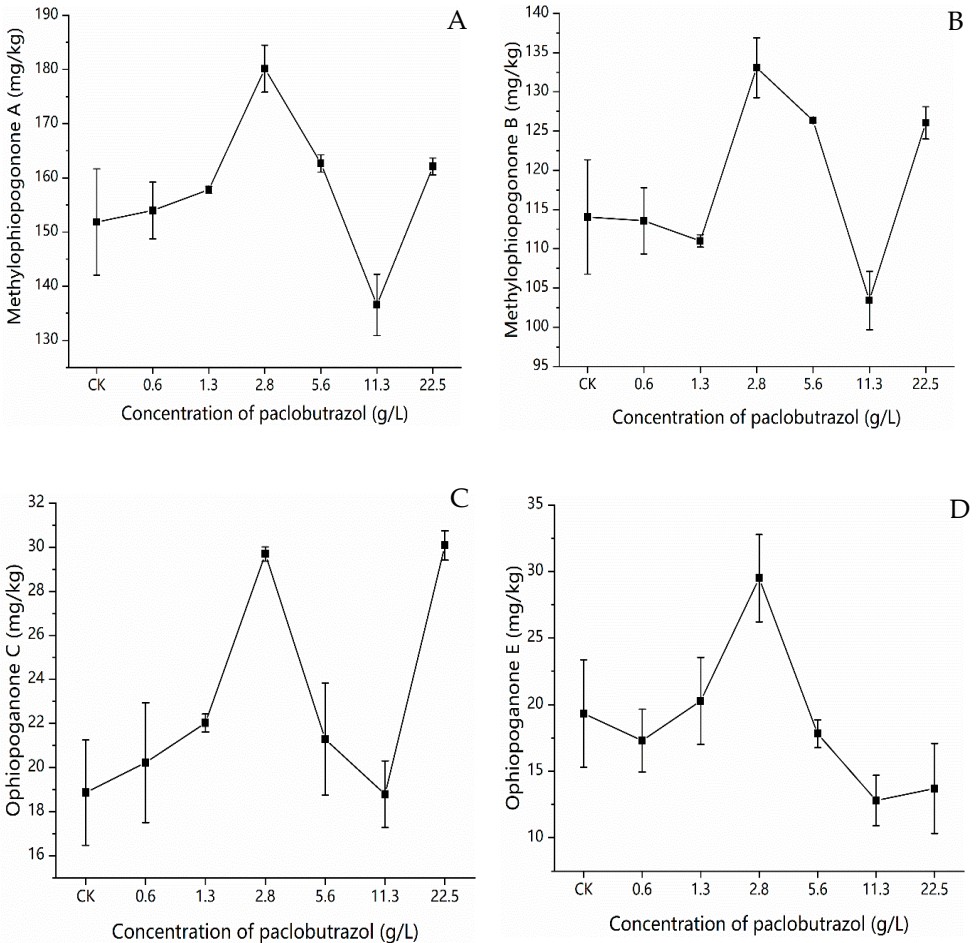

**Figure 8.** Effects of different concentrations of paclobutrazol on Methylophiopogonone A (**A**), Methylophiopogonone A (**B**), Ophiopoganone C (**C**) and Ophiopoganone E (**D**) content. Vertical bars represent standard errors.

## 4. Conclusions

Our study explored the effects of different concentrations of paclobutrazol on the physiological and biochemical effects of *O. japonicus*. Due to the lack of policies and regulations, there is no current regulation regarding the residue limit of *O. japonicus*, which leads to farmers misunderstanding how to apply paclobutrazol rationally; this indirectly caused the phenomenon of excessive pesticide residues in *O. japonicus*. Our research has set a total of six application concentrations; when the application concentration exceeded 11.3 g/L, the final pesticide residues will exceed 0.5 mg/kg. In order to explore the most suitable paclobutrazol application concentration, we also found that even a low

concentration has a significant impact on the growth of *O. japonicus* plants. At medium and low concentrations (0.6 to 11.3 g/L), it was found that an increase in paclobutrazol concentration inhibited the growth of the ground part of *O. japonicus*, and the growth-promoting effect of tuber roots improved. High concentrations of paclobutrazol (22.5 g/L) had a significantly weakened inhibitory effect in height and growth-promoting effect in tuber roots. The same trend was also observed in the 100 SW. The reason for this trend may be the pharmacological effects of high concentration of paclobutrazol. Although high concentrations of paclobutrazol can increase root weight, the maximum weight gain effect was reached at a concentration of 11.3 g/L, which already exceeds 0.5 mg/kg. In addition, the application of paclobutrazol also had an effect on the cell morphology of *O. japonicus*, causing their cell wall to thicken, making the cells smaller and densely arranged. Compared with the control group, paclobutrazol caused severe cell wall separation and organelle loss. However, we found that paclobutrazol has a good inhibitory effect on bacterial growth, which was inversely correlated with its concentration. LC-MS/MS were used to quantitatively analyze the saponins and flavonoid in *O. japonicus*, and the results show that 1.3 g/L paclobutrazol can maximize the accumulation of saponins, and 2.8 g/L paclobutrazol can increase flavonoid content the most.

Therefore, we need to treat the use of paclobutrazol more comprehensively and guide farmers to use pesticides rationally. We found that although the application of paclobutrazol at 11.3 g/L can maximize the growth of *O. japonicus* roots, it also seriously affected the accumulation of flavonoids and saponins, leading to a higher residual concentration of paclobutrazol. Comprehensively, it is believed that the application of 2.8 g/L of paclobutrazol can increase yield while promoting the accumulation of saponins and flavonoids, reducing the pesticide application cost of farmers, obtaining the maximum benefit.

**Author Contributions:** Formal analysis, R.L. and Z.Z.; data curation, R.L., Z.Z.; writing—original draft preparation, Z.Z. and R.L.; writing—review and editing, D.C., J.C., Z.K. and X.D.; methodology, D.C., O.X. and Z.K.; validation, D.C. and O.X.; software, R.L. and Z.Z.; investigation, R.L.; supervision, Z.K. and X.D.; project administration, X.D. and Z.K.; funding acquisition, X.D. and Z.K. All authors have read and agreed to the published version of the manuscript.

**Funding:** This study was supported by China Agriculture Research System of MOF and MARA, Key Project at Central Government Level: The ability establishment of sustainable use for valuable Chinese medicine resources (2060302) and the Beijing Nova Program of Science and Technology (Z191100001119121).

**Institutional Review Board Statement:** Not applicable.

**Informed Consent Statement:** Not applicable.

**Data Availability Statement:** Data sharing does not apply to this article.

**Acknowledgments:** Thank Dan Zhao for their support to the investigation of producing areas and sample collection.

**Conflicts of Interest:** The authors declare no conflict of interest.

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
