# Peer review of "Effect of Paclobutrazol on the Physiology and Biochemistry of Ophiopogon japonicus"

_agronomy, doi:10.3390/agronomy11081533_

Round 1
Reviewer 1 Report
This paper presents some results that will be useful to the growers who are using this chemical to produce Ophiopogon japonicas. Optimum use os this chemical as suggested in this paper will not only increase yield and profit but will also reduce residual impact of this chemical on the health of consumers and the environment.
I think this paper is worth publishing. However, I have few comments and edits that need to be addressed by the authors. These are mentioned in the following Table:
|
Page/ Line |
Edits |
Comments |
|||||||||||||||||||||
|
2/55 |
Replace ‘2’ by ‘two’ |
Here and elsewhere |
|||||||||||||||||||||
|
2/61 |
Ground?? Is this above-ground or below-aground or both Above– and below-ground? |
Be specific |
|||||||||||||||||||||
|
3/102 |
Replace 7 by seven |
Here and elsewhere |
|||||||||||||||||||||
|
4/163 |
Italicise O. japonica |
|
|||||||||||||||||||||
|
Figure 1 |
The zero (untreated control): you have mentioned ‘0” in the text but Al on a sudden introduced as CK in the Figure! But there is no explanation of ‘CK’ anywhere else!!! |
Fix it |
|||||||||||||||||||||
|
9/282 |
Weight? Is this root or shoot weight? Or both? |
Fix it This paper presents some results that will be useful to the growers who are using this chemical to produce Ophiopogon japonicas. Optimum use os this chemical as suggested in this paper will not only increase yield and profit but will also reduce residual impact of this chemical on the health of consumers and the environment. I think this paper is worth publishing. However, I have few comments and edits that need to be addressed by the authors. These are mentioned in the following Table:
|
Author Response
Thank you for your comment
Please see the attachment

Reviewer 2 Report
The work presented for review is quite interesting, and the results can help plantation and have a real impact on increasing yields. However, requires correction before publication. Below are my comments. I hope the authors consider incorporating them into their revision as I strongly believe that these will improve the quality of the article.
In the introduction more information about the physicochemical properties of Paclobutrazole should be added. Which group of chemical compounds does it belong to? Is the same as tebuconazole, which is registered?
In what amounts is it usually used in cultivation? The information in lines 52-53 is not sufficient to estimate the dose. What are the toxicological parameters of this compound? Its effect on slowing down the root growth may result from the toxic dose. It is necessary to complete it, as it will allow to better understand and explain the phenomenon taking place during the use of this compound.
The headings in Table 1 are incorrect. The ion transitions should be reported and which one was used for quantification and which one was used as confirmation.
In Figure 1, the signature of the Y axis should be corrected instead of g/L should be mg/kg.
Line 225: The sentence should be completed.
The differences between Figure 2 and 4 can be observed. At a concentration of 5.6 g/L the root system is longer than in the cases of other doses. Can you comment on this?
Lines 287, 311, 313, 315, 317, 322 – citations need to be corrected.
The abbreviations used should be explained, because the reader has to guess what they mean.
Figure 7 is duplicated.
Very surprising are big differences (jumps) concentrations in Figure 8. Are their content is only dependent on the concentration of added agent or something else affects their level?
Author Response

(The authors gave the same response as above.)
